# Machinery and Developmental Role of Retinoic Acid Signaling in Echinoderms

**DOI:** 10.3390/cells11030523

**Published:** 2022-02-02

**Authors:** Shumpei Yamakawa, Hiroshi Wada

**Affiliations:** Graduate School of Life and Environmental Sciences, University of Tsukuba, Ibaraki 305-8572, Japan; hw98champ@icloud.com

**Keywords:** echinoderms, retinoic acid signaling, metamorphosis

## Abstract

Although a recent genomic survey revealed its ancient evolutionary origin in the animal kingdom, retinoic acid (RA) signaling was previously thought to be unique to chordates. Echinoderms are of critical interest in researching the evolutionary history of RA signaling, as they represent a basal group of deuterostomes. Furthermore, our previous works have suggested that echinoderms may possess the ancestral function of RA signaling for metamorphosis regulation. In this paper, to facilitate future studies of RA signaling in echinoderms, we provide an overview of RA machinery in echinoderms, identify its signaling components, and discuss its developmental role.

## 1. Introduction

In vertebrates, retinoic acid (RA) is synthesized by retinal dehydrogenase (RALDH) and functions by binding to a nuclear retinoic acid receptor (RAR) and a retinoid x receptor (RXR) [1,2]. RAR and RXR form a heterodimer that regulates downstream genes by binding to particular DNA sequences (RARE; retinoic acid response elements) [1,2]. RA signaling plays an essential role in vertebrate developmental processes, for example the patterning of the central nervous system [1,3]. The RA signaling pathway was previously thought to be unique to chordates, as signaling-related genes such RAR have been identified only in this taxon [4,5]; however, several comparative genomic surveys have shown that non-chordate invertebrates also possess the components of RA signaling [5,6,7]. Nonetheless, the mechanism by which RA signaling functions and is conserved in non-chordate invertebrates remains poorly understood.

Echinoderms are of critical interest for understanding the evolution of RA signaling for two key reasons. First, RA signaling plays a primary role in the body patterning of chordates along the anterior–posterior (A–P) axis through the regulation of Hox gene expression [1,2,3], although echinoderms lost a clear A-P axis and now have a unique penta-radial body plan [8]. Second, an ancestral RA signaling function may be conserved in echinoderms. Previously, we reported that RA signaling mediates the metamorphosis process in two starfish species with different developmental modes and the feather star, a basal echinoderm [9,10,11]. RA is also a life cycle transition regulatory molecule in jellyfish, cnidaria [12]. These findings suggest that the regulatory mechanism of life cycle transition is conserved within the bilaterian–cnidarian lineage [9,11].

In this paper, we review the RA signaling machinery and its developmental role in echinoderms and provide a perspective on future research on RA signaling in echinoderms.

## 2. RA Signaling Components in Echinoderms

In vertebrates, retinol is catalyzed to retinal with the usage of retinol dehydrogenase (Figure 1; RDH) [1,2]. RA is synthesized by RALDH and delivered by cellular RA-binding protein (CRABP) into nuclei, where RA binds to the RAR–RXR heterodimer to regulate downstream genes (Figure 1) [1,2]. RA signaling is also regulated by cytochrome P450 26 (CYP26), which degrades endogenous RA (Figure 1) [1,2]. Fatty-acid binding proteins (FABPs) such as FABP5 also modulate RA signaling by transporting RA to peroxisome proliferator-activated receptors (PPAR) that activate a non-canonical signaling pathway (Figure 1) [1]. PPAR makes a heterodimer with RXR binds to specific genomic elements (Figure 1; PPRE, peroxisome proliferator response elements). Based on this information, we examined whether RA signaling components are conserved in echinoderms using public transcriptomic and genomic data (Table 1).

### 2.1. RA Synthesis

#### 2.1.1. Retinol to Retinal

In vertebrates, RDH genes are grouped into two gene families: SDR-RDH (short-chain dehydrogenase/reductase-RDH, e.g., *rdh10*/*sdr16c4*, *rdh11*/*sdr7c1* and *dhrs7*/*retsdr4*) and MDR-ADH (medium-chain alcohol dehydrogenase, e.g., *adh3*, *adh4*) [6]. Both SDR-RDH and MDR-ADH genes have been identified in non-chordate deuterostomes, protostomes and even cnidarians [6,18,19], suggesting that the origin of RDH genes predates before the divergence of cnidarians and bilaterians. For example, a recent study identified the *dhrs7* genes in sea cucumbers and sea urchins [18], and *adh3* genes in protostomes such as annelids [19]. In addition to these identifications, we were able to extract the *dhrs7* and *adh3* genes from all classes of echinoderms (Figure 2). This result indicates the presence of RDH genes in echinoderms, suggesting that the pathway to catalyze retinol to retinal is conserved. Future studies should focus on the comprehensive identification of SDR-RDH and MDR-ADH genes in echinoderms and their ability of retinol oxidation.

#### 2.1.2. Retinal to RA

Vertebrates typically have four *raldh* genes: *raldh1*–*3* (*aldehyde dehydrogenase 1a1*–*3*, *aldh1a1*–*3*) and *raldh4* (*aldh8a1*) [1,23]. The *aldh1a* genes are focused on the research of RA signaling pathway [1], although no *aldh1a* genes have been identified in the sea urchin *Strongylocentrotus purpuratus* [4,6]. By contrast, we previously identified three *aldh1a* genes (*aldh1a-a*, *b*, and *c*) from transcriptome data of the starfish *Patiria pectinifera* and one *aldh**1**a* gene from transcriptome data of the sea lily *Metacrinus rotundus* [9,11]. We also identified one *aldh1a* gene from a genomic dataset of the brittle star *Amphiura filiformis* (Figure 3), indicating that common ancestors of echinoderms had *aldh1a* gene(s). We furthermore found that functional domain residues such as NAD binding sites were well conserved among vertebrate and echinoderm *aldh1a* genes (Appendix A). Notably, no *aldh1a* genes were retrieved from transcriptome or genomic data of the sea cucumber *Apostichopus japonicus*. Regarding that we used the latest genome assembly of the sea urchin *S. purpuratus* (Spur_v5.0, contig N50: 2 Mb, scaffold N50: 37 Mb) and the sea cucumber *A. japonicus* (contig N50: 190 Kb, scaffold N50: 486 Kb) to identify *aldh1a* genes in this study [14], it is likely that *aldh1a* was lost in common ancestors of echinozoans (sea urchins and sea cucumbers).

*aldh8a1* is known as another *raldh* gene in vertebrates [24]. Although no *aldh1a* gene has been retrieved from the genomes of sea urchins and sea cucumbers, Campo-Paysaa et al. [7] identified sea urchin *aldh8a1* genes from genomic data [7]. We also identified *aldh8a1* genes from other echinoderms, including the sea cucumber (Figure 4). Although this finding suggests that RA can be synthesized without *aldh1a* genes in echinozoans, it must be noted that recent work suggest a reassignment of *aldh8a1* function to the kynurenine pathway in tryptophan catabolism [25]. Therefore, although it is likely that RA signaling still functions in the echinoderms including echinozoans, the enzymatic ability for RA synthesis still needs to be demonstrated in future studies.

### 2.2. RA Degradation

RA signaling can be regulated through RA degradation, as CYP26 regulates RA signaling by degradation of RA to develop an RA gradient for morphogenesis [1,26,27]. Vertebrates have three *cyp26* genes showing differential expression: *cyp26a1, cyp26b1*, and *cyp26c1* [1,26,27]. These genes perform crucial roles in various biological processes such as retina formation [1]. The *cyp26* gene was identified in sea urchins [4], with a single *cyp26* gene each identified in transcriptome data of the starfish *P. pectinifera*, the brittle star *A. filiformis*, and the sea lily *M. rotundus*, and two genes identified in the sea cucumber *A. japonicus* (Figure 5). Common ancestors of echinoderms had the c*yp26* gene(s). Note that we also found that the functional domain residues such as heme binding sites were well conserved among human and echinoderm *cyp26* genes (Appendix A).

### 2.3. RA Transport

Cellular RA transport, which involves CRABP, an intercellular lipid binding protein (iLBP) in the same family as FABPs and retinol binding proteins (RBPs), is another regulatory mechanism [1,28]. Previous studies have indicated that several *fabp-like* genes have RA transportation capacity in protostomes, although their orthology with vertebrate CRABP genes is not supported [29,30]. In deuterostomes, Orito et al. [31] classified vertebrate FABPs into three subfamilies and identified the basal gene(s) of each subfamily using urochordate genomic data; however, they did not identify any genes that were orthologous with the *crabp* and *rbp* genes, suggesting that CRABPs are components specific to vertebrates [31]. By contrast, based on sea urchin genomic data, one *crabp* gene, two *rbp* genes (*rbp-1* and *rbp-2*), and two *fabp* genes (*fabp-1* and *fabp-2)* were identified [32], although no phylogenetic analysis has been conducted.

A phylogenetic tree of iLBP family genes from chordates, echinoderms, and protostomes was constructed (Figure 6). Based on the transcriptome/genomic data of other echinoderms, several genes that form respective clades with sea urchin *crabp*, *rbp*, and *fabp* genes were identified (Figure 6). However, the phylogenetic relationship of these clades with the vertebrate genes could not be resolved. Considering that vertebrate *rbp* and *crabp* genes formed a clade without invertebrate genes, three echinoderm clades contained rather similar *fabp* genes, suggesting that these should be renamed as subfamilies *fabp-like* 1, *2*, and *3*, respectively. Therefore, although the evolutionary history of *fabp* (like) genes remains unclear, these results suggest that RA transport by CRABP is a trait specific to vertebrates.

### 2.4. RA Receptors

In vertebrates, RA functions by binding with the RAR–RXR heterodimer, the components of which are encoded by three genes in most vertebrate [1,2]. RAR and RXR primarily bind with all-trans RA and 9-cis RA, respectively, although selective binding of RAR with all-trans RA is particularly important for signal transduction [1,2]. Although a single *rar* gene and a unique *rxr* gene were identified in all classes of echinoderms (Figure 7), it has not been experimentally demonstrated that the RAR-RXR heterodimer regulates the expression of downstream genes in the same manner as in vertebrates. Whether RAR selectively binds with all-trans RA in echinoderms also remains unclear.

Insight into the conservation of the RA receptor function can be obtained from protostomes. Handberg-Thorsager et al. [19] reported that RAR forms a heterodimer with RXR to bind all-trans RA in the annelid *Platynereis dumerilii* [19]. They also revealed that a chimeric construct with the ligand binding domain (LBD) of RAR fused to the GAL4 domain is able to activate a reporter gene in vitro in presence of RA [19]. Moreover, Fonseca et al. [33] demonstrated that the LBD of RAR can induce a reporter gene in the priapulid *Priapulus caudatus* [33]. These findings are not directly applicable to echinoderms but suggest that the machinery of the RAR–RXR heterodimer is conserved throughout bilaterians that have *rar* and *rxr* genes.

The two studies mentioned above also shed light on the affinity of ligand binding of RAR. Handberg-Thorsager et al. [19] revealed that the RAR of the annelid *P. dumerilii* shows a lower ligand binding affinity than the RAR of chordates [19]. Fonseca et al. [33] found that the ligand binding affinity of RAR is much lower in the priapulid *P. caudatus* than in the annelid *P. dumerilii* [33]. These reports suggested that the ancestral RAR is low-affinity sensor for RA. Handberg-Thorsager et al. [19] discussed that the ligand binding affinity of sea urchin RAR is comparable to the annelid *P. dumerilii* RAR [19]. In addition to these reports, this study also found that the echinoderm RAR showed several mismatch-residue in their ligand binding pockets with the human RAR (Appendix A). It is interesting to determine the ligand binding affinity of echinoderm RARs.

### 2.5. Regulation of Downstream Genes

The RAR–RXR heterodimer recognizes a specific DNA sequence, the RA response element (RARE), in the promoter or enhancer regions of downstream genes [1,34,35]. RARE is typically composed of two direct repeats (DRs) with a conserved nucleotide sequence (A/G)G(G/T)TCA [36]. This sequence is further separated by a spacer of one, two or five nucleotides (respective elements are termed as DR1, DR2, and DR5 elements) [36]. The recent genomic survey revealed that RAREs position in the promoter regions of the downstream genes of RA signaling such as Hox genes in chordates [1].

In echinoderms, Marlétaz et al. [3] found a putative DR5 RARE at the 3770 bp 5′-upstream of the sea urchin *hox1* gene [3]. Carvalho et al. [36] also identified the D2 RARE (AGTTCAATAGTTCA) at the 2503 bp 5′-upstream of the sea urchin *cyp26* gene [36]. Despite these reports, the presence of RAREs have not investigated in other echinoderm genomic data. Furthermore, it is still unclear if RAR/RXR heterodimer specifically recognizes the RARE in the genome of echinoderm.

### 2.6. Non-Canonical Signaling Pathway

Schug et al. [37] reported that PPARb/g activated by RA binding repressed apoptosis in mammalian cell lines, whereas RA binding to RAR promoted the same process [37]. Furthermore, FABP5 binds and transports RA to PPARb/g, and CRABP delivers RA to RAR [37]. These results suggest that cell growth is affected by the proportions of FABP5 and CRABP [37].

This signaling pathway, activated by FABP and PPAR, is considered a “non-canonical” RA signaling pathway [1]. In the present study, *ppar* genes were found in each class of echinoderm (Figure 7). However, because echinoderm *fabp-like* genes do not appear to be orthologous to vertebrate *fabp* genes, as mentioned above, it remains unclear whether the non-canonical pathway functions in echinoderms. Otherwise, considering that signaling with PPAR performs various functions in vertebrates (e.g., in cellular metabolism) [38], PPAR in echinoderms would likely perform a function other than mediating the RA non-canonical signaling pathway.

## 3. Developmental Role of RA Signaling in Echinoderms

Based on the phylogenetic analysis, we suggest that, despite the absence of some signaling components, similar machineries are used to regulate RA signaling in echinoderms (Figure 8). Next, we summarize previous research on the developmental roles of RA signaling in echinoderms to provide a perspective on future studies of RA signaling functions in echinoderms.

### 3.1. Exogenous RA Treatment

Several studies have examined the effects of exogenous RA on echinoderm development. First, Sciarrino and Matranga [39] reported that, with the exception of delaying development, RA treatment did not induce a specific phenotype in the sea urchin *Paracentrotus lividus* [39]. In addition, pseudopodial cable growth on micromere-delivered cells was reported in the sea urchin *Hemicentrotus pulcherrimus* following RA treatment [40]. However, these results should be re-examined using expression analyses of signaling component genes such *aldh1a*, *rar*, *rxr* and *cyp26*.

### 3.2. Metamorphosis Regulation

#### 3.2.1. RA Signaling Is Involved in the Regulation of Starfish Metamorphosis

Yamakawa et al. [9] combined pharmacological and gene expression analyses to demonstrate that RA signaling is involved in metamorphosis regulation in the starfish *P. pectinifera* [9]. In the gain-of-function experiment, exogenous RA treatment induced metamorphosis in starfish larvae at the competent stage for metamorphosis [9]. By contrast, the loss-of-function by inhibitors of RA synthesis and RA receptors suppressed metamorphosis triggered by attachment to a substrate via brachiolar arms [9]. We found that *aldh1a-a, b, c*, *rar*, and *rxr* were expressed in juvenile rudiments of competent or metamorphosing larvae, suggesting that RA signaling is activated to commence metamorphosis after larvae settle onto a substrate [9].

The function of RA signaling in metamorphosis regulation was also reported in a different starfish species, *Astropecten latespinosus*, which subsequently lost its feeding/digestion system and brachiolar arms in larval forms [10]. We found that the metamorphosis was induced by providing natural sand from the adult habitat, suggesting that *A. latespinosus* larvae sense specific environmental cues for metamorphosis [10]. We also found that exogenous RA treatment induced metamorphosis in competent larvae, the inhibition of RA synthesis or the binding of a RAR antagonist (RO41-5253) suppressed metamorphosis, and RA signaling-related genes were expressed in the juvenile rudiment [10]. Therefore, RA signaling may mediate metamorphosis commencement upon the reception of environmental cues, even in *A. latespinosus*, which shows derived larval forms [10].

We note that the *aldh1a*, *rar*, and *rxr* genes are already expressed in the floating larval stages of both species, before settlement and metamorphosis [9,10]. Therefore, the mechanism underlying the timing of metamorphosis regulation remains unclear. Future studies should explore why RA signaling is not activated in competent larvae despite the expression of *aldh1a*, *rar*, and *rxr*, as well as the mechanism by which RA signaling is activated by stimuli received via sensory organs (brachiolar arms in *P. pectinifera* and unidentified organs in *A. latespinosus*).

#### 3.2.2. Ancestral Function of RA Signaling in Echinoderms

Fuchs et al. [12] reported that medusozoan life-cycle transition is stimulated by endogenous RA, and suggested that 9-cis RA binds to RXR with another nuclear receptor to regulate downstream genes in cnidarians lacking the *rar* gene [12]. Thus, RA signaling functions differently in cnidarians than in starfish, although the present findings suggest that the component of RA signaling that regulates metamorphosis in starfish was potentially involved in life cycle regulation in common ancestors of cnidarians and bilaterians.

In this context, it is important to determine the ancestral function of RA in echinoderms. Therefore, we investigated the role of RA in the metamorphosis of the feather star (crinoid), a basal group of echinoderms [11]. We found that treatment with exogenous RA induced metamorphosis processes such as development of the calyx, stalk, and adhesive pit in competent doliolaria larvae of the feather star *Antedon serrata* [11]. By contrast, pharmacological inhibition of RA synthesis or binding to RAR suppressed metamorphosis after settlement [11]. These results suggest that RA signaling regulates feather star metamorphosis, supporting the regulation of metamorphosis by RA signaling in the common ancestor of living echinoderms [11].

Despite these findings, the function of RA in life-cycle transition has been reported only in starfish, feather stars, and jellyfish (Figure 9) [9,10,11,12]. For example, a recent study using the annelid *Platynereis dumerilii* reported that RA signaling functions as a sensor for neurogenesis in trochophore larvae, although RA signaling involvement in metamorphosis remains unclear [27]. In the deuterostomes, Sasakura et al. 2012 reported that RA signaling mediates the expression of *hox1* which is required for the epidermal otic/atrial placodes formation during metamorphosis of the ascidian *Ciona intestinalis* [41]. However, RA signaling has not been reported to be involved in the metamorphosis regulation of sea urchins or ascidians [42,43], deuterostomes among which molecular metamorphosis regulation has been investigated in detail. Therefore, further investigations using various animal taxa are required to support our hypothesis.

### 3.3. Insight into the Evolution of RA Signaling Function for Chordate Hox Gene Expression Regulation

Although the hypothesis of an ancestral function of RA signaling in metamorphosis remains to be tested, it carries further implications for the evolution of RA signaling. RA signaling may have been co-opted for the patterning of the chordate body plan via the recruitment of Hox genes as regulators. It is an interesting coincidence that echinoderm Hox genes show collinear expression during stages when larvae are becoming competent for metamorphosis [44]. Thus, RA signaling may be involved in the collinear regulation of Hox genes. Although no evidence for RA regulation of Hox genes has been reported in extant echinoderms, this lack of data does not necessarily indicate that RA did not regulate Hox gene expression in ancestral deuterostomes. One group of chordates, the larvaceans, lost the regulation of Hox genes by RA [45], although substantial evidence supports the presence of the regulation of Hox genes by RA in common ancestors of the chordates [3]. Our understanding of echinoderm RA signaling is probably far from comprehensive. Further study of RA signaling in other echinoderms may elucidate the contribution of RA signaling to echinoderm and deuterostome evolution.

### 3.4. Regeneration

We finally want to note another aspect of RA signaling as a regeneration regulator. The regulatory function of RA signaling has been reported in the regeneration process of various tissues and organs of vertebrates including fish, amphibians and mammals [46,47,48]. For example, the expression of *aldh1a* gene is induced in the reaction to damage in the zebrafish heart and fin [46]. This expression causes local RA synthesis to activate regeneration process such as cell proliferation [46]. Based on these findings, RA has been considered as a regeneration-inducing molecule in vertebrates [48].

Recently, it was revealed that RA signaling is also involved in the regeneration of echinoderms [18,49,50,51]. Viera-Vera and Garcia-Arrars 2018, 2019 focused on the digestive regeneration of the sea cucumber *Holothuria glaberrima* and investigated the involvement of RA signaling through gene expression and pharmacological analysis [18,49]. They found that RA signaling machinery including *rar* differently expressed during regeneration [49]. Furthermore, the size of regenerating intestinal rudiment decreased by the treatment of RAR antagonist (LE135) or RA synthesis inhibitor (Citral) [18]. This data suggests that RA signaling performs roles of modulating cellular dedifferentiation and division that are required for the intestinal regeneration in sea cucumbers [18]. Although other echinoderms and invertebrates show the ability for regeneration [52], the RA signaling function has been fragmentally investigated. Therefore, it is interesting to investigate if the role of RA signaling in regeneration is evolutionary conserved among an animal kingdom.

## Figures and Tables

**Figure 1 cells-11-00523-f001:**
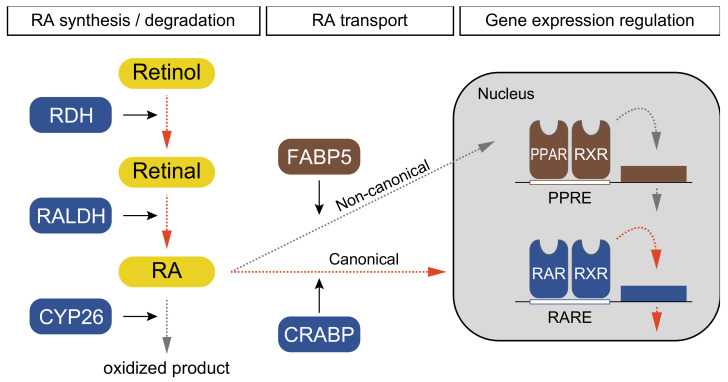
Schematic pathway and components of retinoic acid (RA) signaling in echinoderms. The signaling pathway is divided into three steps: RA synthesis/degradation, RA transport, and gene expression regulation. Dotted arrows indicate the direction of signaling transduction; orange and gray indicate canonical and non-canonical pathways, respectively. Squares indicate enzymes and retinal derivatives; blue, brown indicate enzymes with canonical and non-canonical pathways, respectively, and yellow indicates retinol derivatives. Solid arrows indicate enzyme functions.

**Figure 2 cells-11-00523-f002:**
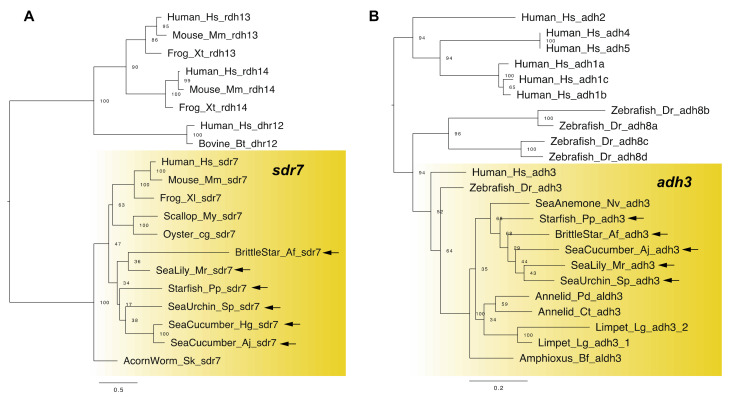
Maximum-likelihood phylogenetic tree for RDH genes (**A**: *sdr7* and **B**: *adh3*). All sequences are presented in Appendix A. Sequences were aligned using the MAFFT v7 software with the default parameters [20]. Amino acid sites for tree construction were selected using the trimAl (1.2rev59) tool with a gap threshold of 0.8 [21]. A best-fitting amino acid substitution model and a maximum-likelihood tree were inferred using the RaxML v8.2.12 software [22]. Confidence values shown at nodes were calculated after 1000 bootstrap runs. Yellow box and arrows, respectively, indicate *sdr7* (**A**), *adh3* (**B**) gene clade and the genes retrieved from dataset of echinoderms. Abbreviation: Human; *Homo sapiens*, Mouse; *Mus musculus*, Bovine; *Bos taurus*, Frog; *Xenopus tropicalis*, Frog; *Xenopus leavis*, Zebrafish; *Danio rerio*, Amphioxus; *Branchiostoma floridae*, Acorn worm; *Saccoglossus kowalevskii*, Sea cucumber; *Holothuria glaberrima*, Scallop; *Mizuhopecten yessoensis*, Oyster; *Crassostrea gigas*, Annelid; *Platynereis dumerilii*, Annelid; *Capitella teleta*, Limpet; *Lottia gigantea* and Sea anemone; *Nematostella vectensis*.

**Figure 3 cells-11-00523-f003:**
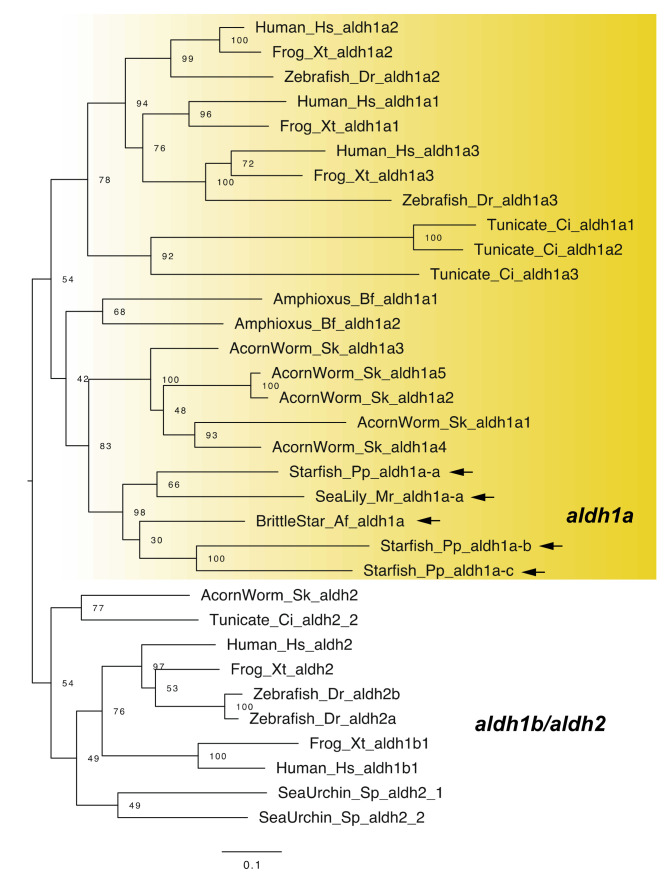
Maximum-likelihood phylogenetic tree for *aldh1a*, *-1b* and *2* genes. We retrieved the gene dataset of *aldh1a*, *-1b* and *2* (Appendix A) and constructed phylogenetic tree as described above. Yellow box and arrows, respectively, indicate *aldha1* gene clade and *aldha1* genes retrieved from dataset of echinoderms. Abbreviation: Tunicate; *Ciona intestinalis*.

**Figure 4 cells-11-00523-f004:**
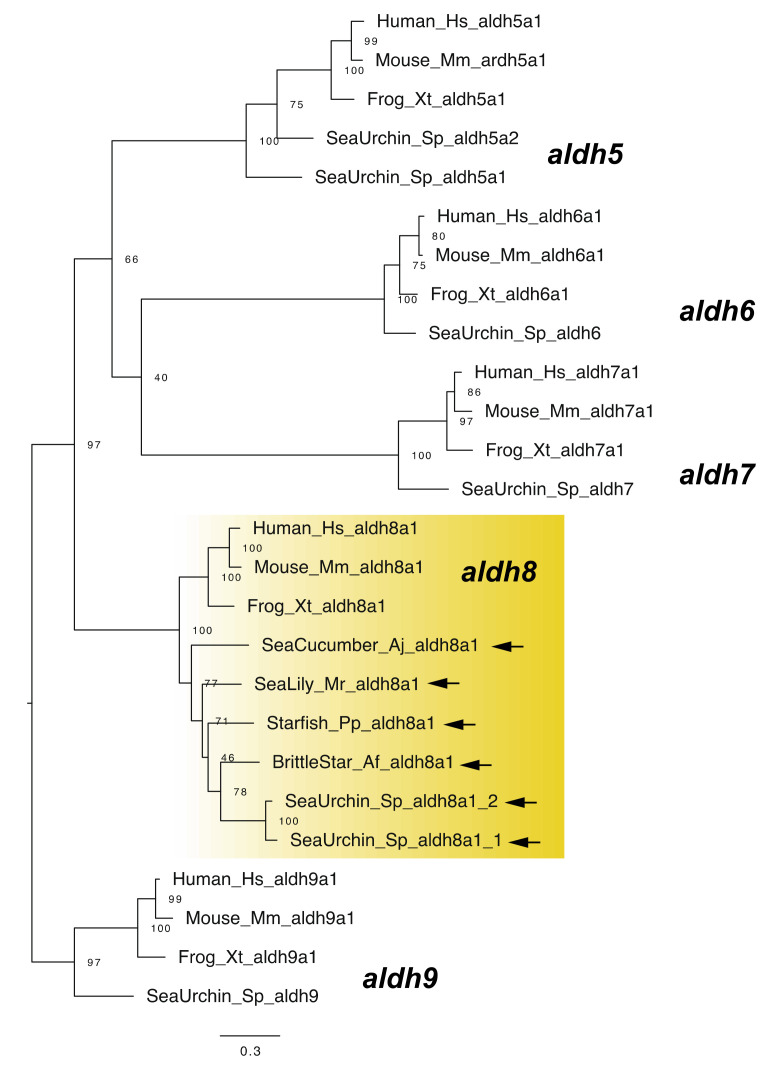
Maximum-likelihood phylogenetic tree for *aldh5*–*9* genes. We retrieved the gene dataset of *aldh5*–*9* (Appendix A) and constructed phylogenetic tree as described above. Yellow box and arrows, respectively, indicate *aldh8a1* gene clade and *aldh8a1* genes retrieved from dataset of echinoderms.

**Figure 5 cells-11-00523-f005:**
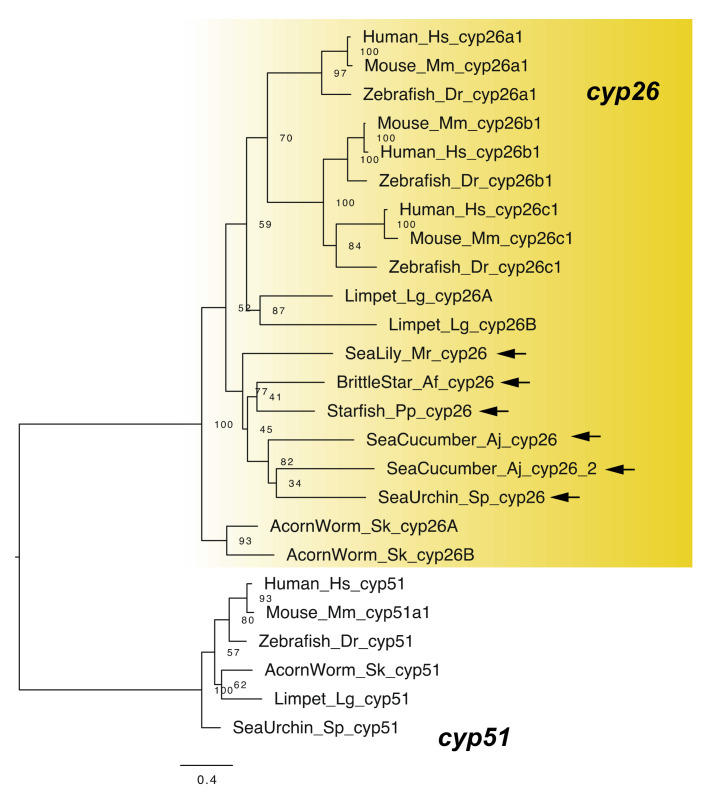
Maximum-likelihood phylogenetic tree for *cyp* genes. We retrieved the gene dataset of *cyp* (Appendix A) and constructed phylogenetic tree as described above. In addition, construct phylogenetic tree, respectively. Yellow box and arrows, respectively, indicate *cyp26* gene clade and *cyp26* genes retrieved from dataset of echinoderms.

**Figure 6 cells-11-00523-f006:**
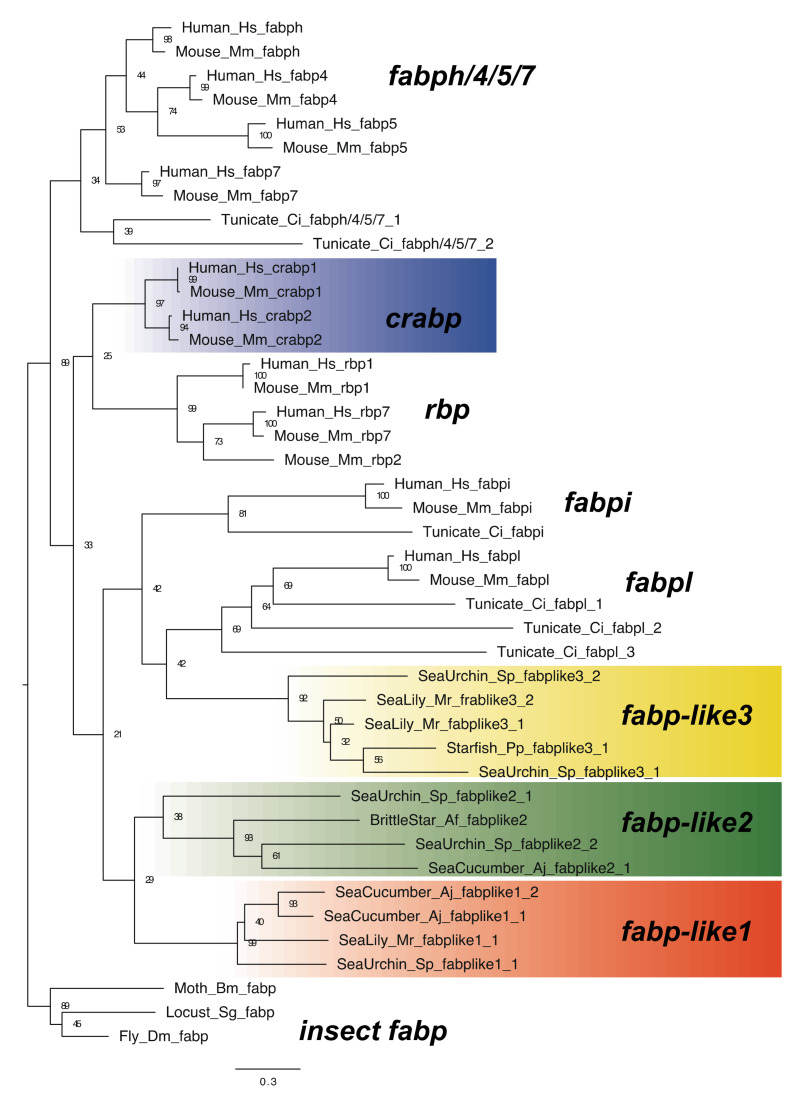
Maximum-likelihood phylogenetic tree for iLBP genes. We retrieved the dataset of iLBP genes (Appendix A) and constructed phylogenetic tree as described above. Each color box indicates the gene clade of *crabp* (blue), *fabp-like1* (orange), *fabp-like2* (green) and *fabp-like3* (yellow). Abbreviation: Fly; *Drosophila melanogaster*, Locust; *Schistocerca gregaria* and Moth; *Bombyx mori*.

**Figure 7 cells-11-00523-f007:**
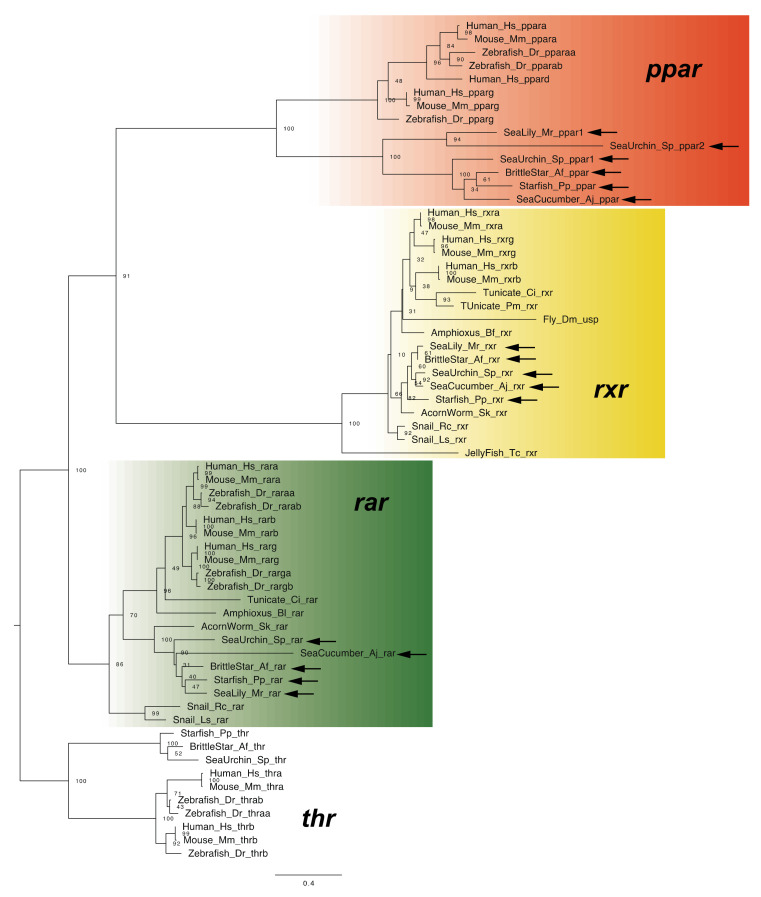
Maximum-likelihood phylogenetic tree for nuclear receptor genes. We retrieved the dataset of nuclear receptor genes (Appendix A) and constructed phylogenetic tree as described above. Each color box indicates the gene clade of *ppar* (**orange**), *rar* (**green**) and *rxr* (**yellow**). Arrows indicate the genes retrieved from dataset of echinoderms. Abbreviation: *thr*; *thyroid hormone receptor*, Amphioxus (Bl); *Branchiostoma lanceolatum*, Tunicate (Pm); *Polyandrocarpa misakiensis*, Snail (Rc); *Reishia clavigera*, Snail (Ls); *Lymnaea stagnalis* and Jellyfish; *Tripedalia cystophora*.

**Figure 8 cells-11-00523-f008:**
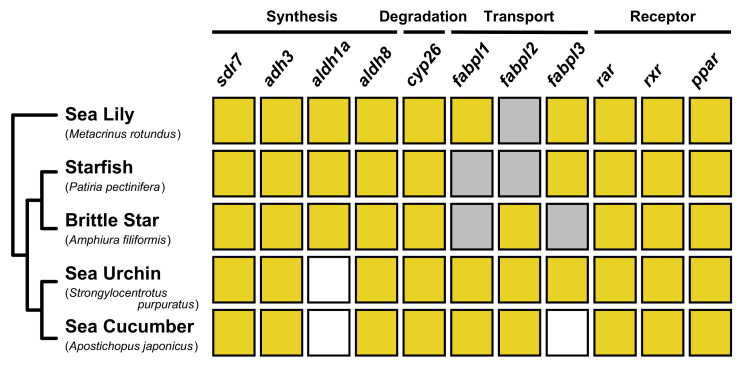
Summary of gene identification of RA signaling components in echinoderms. Yellow boxes indicate the presence of ortholog genes. On the other hand, gray and white boxes indicate the absence of ortholog genes in the search of transcriptomic and genomic dataset, respectively.

**Figure 9 cells-11-00523-f009:**
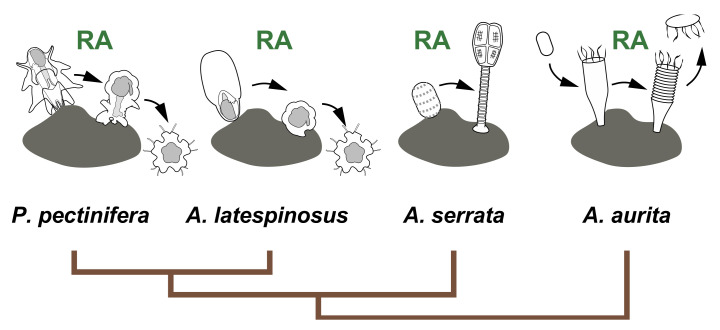
The function of RA signaling in life cycle transition of echinoderms and cnidarians. The involvement of RA signaling in metamorphosis is reported in the starfish *P. pectinifera* [9], *A. latespinosus* [10] and the feather star *A. serrata* [11]. RA is also known as the regulator of strobilation in the jellyfish *Aurelia aurita* [12].

**Table 1 cells-11-00523-t001:** Resource of genomic and transcriptomic data of echinoderms. We investigated transcriptome data for the starfish *Patiria pectinifera* [9], the sea cucumber *Apostichopus japonicus* [13], the brittle star *Amphiura filiformis* [14] and the sea lily *Metacrinus rotundus* [15]. We also used genomic data for the sea urchin *Strongylocentrotus purpuratus* [16], and the sea cucumber *A. japonicus* (http://www.genedatabase.cn/aja_genome_20161129.html, accessed on 28 January 2022) [17]. Sequences for other animals were obtained from Uniprot (https://www.uniprot.org/, accessed on 28 January 2022) and GenBank (https://www.ncbi.nlm.nih.gov/genbank/, accessed on 28 January 2022).

Class	Species	Data Type	Resource
Crinoid/Sea lily	*Metacrinus rotundus*	Transcriptome	Koga et al., 2016 [15]
Asteroid/Starfish	*Patiria pectinifera*	Transcriptome	Yamakawa et al., 2018 [9]
Ophiurida/Brittle star	*Amphiura filiformis*	Transcriptome	Dylus et al., 2018 [14] echinonet (http://echinonet.org.uk/, accessed on 28 Janaury 2022)
Echinoid/Sea urchin	*Strongylocentrotus purpuratus*	Genome	Spur_v3.1, Spur_v5.0 (the version of assembly)
			Echinobase [16]
Holothuroid/Sea cucumber	*Apostichopus japonicus*	Transcriptome	Jo et al., 2007 [13]
		Genome	Zhang et al., 2017 [17] The Sea Cucumber *Apostichopus japonicus* Genome (http://www.genedatabase.cn/aja_genome_20161129.html, accessed on 28 Janaury 2022)

## Data Availability

All sequences which were used in the phylogenetic analysis were deposited in the Appendix A.

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
