# Peer review of "Machinery and Developmental Role of Retinoic Acid Signaling in Echinoderms"

_cells, 2022, doi:10.3390/cells11030523_

Round 1

Reviewer 1 Report

This manuscript is well written and easy to follow. My only minor comment is about the participating of RA in another developmental process, regeneration. It is well known that echinoderms have good regenerative ability. There are some publications about expression of RA signaling pathway during regeneration in the animals. You can find the papers in review of Dolmatov IY. Cells, 2021, 12, 250. I think authors need to add short chapter about the role of RA pathway in echinoderm regeneration, especially since there is a similarity with vertebrates, in which the RA pathway is activated during regeneration

Another remarks:

  1. Legend of Fig. 3.

Yellow box and arrows respectively indicate aldh8 gene clade and aldh8 genes, but not aldh5.

  1. Lines 94-101, and below in the text

Gene names and Latin names of animals should be written in italic.

  1. Fig5, clade fabp-like3 (yellow)

“Sealily_Mr_frablike3_2_2” – maybe “Sealily_Mr_fabplike3_2_2”?

  1. Fig. 6 is in low quality. It needs to be changed by fig with better resolution/quality.

Author Response

Reviewer 1

This manuscript is well written and easy to follow. My only minor comment is about the participating of RA in another developmental process, regeneration. It is well known that echinoderms have good regenerative ability. There are some publications about expression of RA signaling pathway during regeneration in the animals. You can find the papers in review of Dolmatov IY. Cells, 2021, 12, 250. I think authors need to add short chapter about the role of RA pathway in echinoderm regeneration, especially since there is a similarity with vertebrates, in which the RA pathway is activated during regeneration

> According to this comment, we reviewed the reports about the role of RA pathway in echinoderm regeneration as the following.

L328–348

3.4 Regeneration

We finally want to note another aspect of RA signaling as a regeneration regulator. The regulatory function of RA signaling has been reported in the regeneration process of various tissues and organs of vertebrates including fish, amphibians and mammals [46–48]. For example, the expression of aldh1a gene is induced in the reaction to damage in the zebrafish heat and fin [46]. This expression causes local RA synthesis to activate regen-eration process like cell proliferation [46]. Based on these findings, RA has been con-sidered as a regeneration-inducing molecule in vertebrates [48].

Recently, it was revealed that RA signaling is also involved in the regeneration of echinoderms [18,49–51]. Viera-Vera and Garcia-Arrars 2018, 2019 focused on the digestive regeneration of the sea cucumber Holothuria glaberrima and investigated the involvement of RA signaling through gene expression and pharmacological analysis [18,49]. They found that RA signaling machinery including rar differently expressed during regen-eration [49]. Furthermore, the size of regenerating intestinal rudiment decreased by the treatment of RAR antagonist (LE135) or RA synthesis inhibitor (Citral) [18]. This data suggests that RA signaling performs roles of modulating cellular dedifferentiation and division that are required for the intestinal regeneration in sea cucumbers [18]. Although other echinoderms and invertebrates show the ability for regeneration [52], the RA signaling function has been fragmentally investigated. Therefore, it is interesting to investigate if the role of RA signaling in regeneration is evolutionary conserved among an animal kingdom.

Another remarks:

Legend of Fig. 3.

Yellow box and arrows respectively indicate aldh8 gene clade and aldh8 genes, but not aldh5.

> Modified accordingly.

Lines 94-101, and below in the text

Gene names and Latin names of animals should be written in italic.

> Modified accordingly.

Fig5, clade fabp-like3 (yellow)

“Sealily_Mr_frablike3_2_2” – maybe “Sealily_Mr_fabplike3_2_2”?

> Modified accordingly.

Fig. 6 is in low quality. It needs to be changed by fig with better resolution/quality.

> Modified accordingly.

Reviewer 2 Report

Cells manuscript review (#1559205)

Machinery and Developmental Role of Retinoic Acid Signaling in Echinoderms

  1. Yamakawa and H. Wada

In this manuscript, Yamakawa and Wada focus on reviewing the genomic, transcriptomic and functional evidence supporting the presence of an active retinoic acid signaling pathway in Echinoderms. Based on sequence homology to genomic or transcriptomic data, the authors assign the presence of a specific retinoic acid network component to a specific species. The authors then go on to review the functional evidence supporting the role of retinoic acid signaling in the life cycle of echinoderms.

This review is important from an evolutionary stand point to understand the establishment of retinoic acid signaling as a fully functional pathway. I have several general comments on the manuscript that require attention before the manuscript can be accepted for publication.

1. As researchers working in echinoderms, I can understand the authors want to convince the reader that retinoic acid signaling is functional in these species but the lack of a critical discussion is lacking. An example would be the conservation in echinoderms of an aldh1a gene. First, the gene nomenclature gets extremely confusing. Comparing the aldh1a genes from vertebrates to sequences/genes named raldha-c, or raldh1a1-3, or raldh2, or raldh2a-b, or other combinations make it extremely difficult to reach conclusions.

2. The issue of incomplete presentation continues from aldh1a to aldh8a1. The identification of these gene seems “solve” the problem. Again, the authors do not cite the work supporting the possible role of this enzyme in retinoic acid biosynthesis (PMID: 11007799) or the more recent paper disputing this contention (PMID: 29703752).

3. The fact that no actual enzymatic activity in echinoderms of the enzyme groups is hardly discussed detracts from their identification. The data is all presented as sequence phylogenetic trees and the reader would greatly benefit from some protein/peptide alignments of functional domains that would define specific enzyme sub-families.

4. It is also important to clarify when describing functional data whether experiments are based on gain- or loss-of-function.

5. An even more general comment is the zoological descriptions. This reviewer has some zoological and evolutionary knowledge. When only species names are given but no general names like starfish or sea urchin are mentioned, it required me to search for the species to know what the authors were talking about. It is very important to make the manuscript readable to as general an audience as possible.

6. A review is a great format to summarize data and present hypothesis while refuting others. This review does a good sequence phylogenetic comparisons but lacks in the functional aspects of the genes/transcripts identified. Perhaps a summary table of sea urchins vs starfish vs sea cucumbers, etc. and which genes they have and which don't would give a clearer picture of the conservation of the retinoic acid network in echinoderms

7. The manuscript should be proofread to improve the English.

Author Response

Reviewer 2

Cells manuscript review (#1559205)

Machinery and Developmental Role of Retinoic Acid Signaling in Echinoderms

Yamakawa and H. Wada

In this manuscript, Yamakawa and Wada focus on reviewing the genomic, transcriptomic and functional evidence supporting the presence of an active retinoic acid signaling pathway in Echinoderms. Based on sequence homology to genomic or transcriptomic data, the authors assign the presence of a specific retinoic acid network component to a specific species. The authors then go on to review the functional evidence supporting the role of retinoic acid signaling in the life cycle of echinoderms.

This review is important from an evolutionary stand point to understand the establishment of retinoic acid signaling as a fully functional pathway. I have several general comments on the manuscript that require attention before the manuscript can be accepted for publication.

  1. As researchers working in echinoderms, I can understand the authors want to convince the reader that retinoic acid signaling is functional in these species but the lack of a critical discussion is lacking. An example would be the conservation in echinoderms of an aldh1a gene. First, the gene nomenclature gets extremely confusing. Comparing the aldh1a genes from vertebrates to sequences/genes named raldha-c, or raldh1a1-3, or raldh2, or raldh2a-b, or other combinations make it extremely difficult to reach conclusions.

> We appreciate this author’s comment. Even though the gene names aldh1a1 and aldh2 are correct, we noticed the mistake of using raldh1a1and raldh2 in the diagram of phylogenic tree. We corrected this mistake in the revised manuscript. Furthermore, we modified the gene nomenclature of echinoderm raldhs to clarify the gene orthologues throughout the revised manuscript: raldha, -b and -caldh1a-a, -b and -c.

  1. The issue of incomplete presentation continues from aldh1a to aldh8a1. The identification of these gene seems “solve” the problem. Again, the authors do not cite the work supporting the possible role of this enzyme in retinoic acid biosynthesis (PMID: 11007799) or the more recent paper disputing this contention (PMID: 29703752).

> According to this comment, we modified our manuscript with citing the papers which this reviewer kindly suggested.

Thus, no aldh1a gene has been retrieved from the genomes of these species, although Campo‐Paysaa et al. [7] identified sea urchin aldh8a1 genes from genomic data [7]. We also identified aldh8a1 genes from transcriptome data of other echinoderms, including the sea cucumber (Fig. 3), suggesting that RA can be synthesized without aldh1a genes in echinozoans. Therefore, although echinozoans lost the aldh1a gene secondarily, it is likely that RA signaling still functions in this taxon.

To

Aldh8a1 is known as another raldh gene in vertebrates [24]. Although no aldh1a gene has been retrieved from the genomes of sea urchins and sea cucumbers, Campo‐Paysaa et al. [7] identified sea urchin aldh8a1 genes from genomic data [7]. We also identified aldh8a1 genes from other echinoderms, including the sea cucumber (Fig. 4). Although this finding suggests that RA can be synthesized without aldh1a genes in echinozoans, it must be noted that recent work suggest a reassignment of aldh8a1 function to the kynurenine pathway in tryptophan catabolism [25]. … (L110–118).

  1. The fact that no actual enzymatic activity in echinoderms of the enzyme groups is hardly discussed detracts from their identification. The data is all presented as sequence phylogenetic trees and the reader would greatly benefit from some protein/peptide alignments of functional domains that would define specific enzyme sub-families.

> Among RA signaling genes, the functional domains have been comparatively well investigated in the RA synthesis enzyme (aldh1a), degradation enzyme (cyp16) and receptor (rar). Thus, according to this reviewer, we discussed the enzymatic activity with showing their alignments in the supplementary files of the revised manuscript.

Ex) aldh1a

By contrast, we previously identified three aldh1a genes (aldh1a-a, b, and c) from tran-scriptome data of the starfish Patiria pectinifera and one aldh1a gene from transcriptome data of the sea lily Metacrinus rotundus [9,11]. We also identified one aldh1a gene from a genomic dataset of the brittle star Amphiura filiformis (Fig. 2), indicating that common ancestors of echinoderms had aldh1a gene(s). We furthermore found that functional domain residues like NAD binding sites were well conserved among vertebrate and echinoderm aldh1a genes (Fig. S1). (L102–104).

  1. It is also important to clarify when describing functional data whether experiments are based on gain- or loss-of-function.

> According to this and another reviewer’s comments, we modified to clarify the gain- or loss-of-function in the revised manuscript. The example is the following.

Exogenous RA treatment induced metamorphosis in starfish larvae at the competent stage for metamorphosis [9]. By contrast, inhibitors of RA synthesis and RA receptors suppressed metamorphosis triggered by attachment to a substrate via brachiolar arms.

to

In the gain-of-function experiment, exogenous RA treatment induced metamorphosis in starfish larvae at the competent stage for metamorphosis [9]. By contrast, the loss-of-function by inhibitors of RA synthesis and RA receptors suppressed metamorphosis triggered by attachment to a substrate via brachiolar arms (L250–257).

  1. An even more general comment is the zoological descriptions. This reviewer has some zoological and evolutionary knowledge. When only species names are given but no general names like starfish or sea urchin are mentioned, it required me to search for the species to know what the authors were talking about. It is very important to make the manuscript readable to as general an audience as possible.

> We modified our manuscript to add the general names throughout the revised manuscript.

  1. A review is a great format to summarize data and present hypothesis while refuting others. This review does a good sequence phylogenetic comparisons but lacks in the functional aspects of the genes/transcripts identified. Perhaps a summary table of sea urchins vs starfish vs sea cucumbers, etc. and which genes they have and which don't would give a clearer picture of the conservation of the retinoic acid network in echinoderms

> According to this comment, we added a summary figure of gene identification (Fig. 8) in the revised manuscript.

  1. The manuscript should be proofread to improve the English.

> Modified Accordingly.

Reviewer 2

Cells manuscript review (#1559205)

Machinery and Developmental Role of Retinoic Acid Signaling in Echinoderms

Yamakawa and H. Wada

In this manuscript, Yamakawa and Wada focus on reviewing the genomic, transcriptomic and functional evidence supporting the presence of an active retinoic acid signaling pathway in Echinoderms. Based on sequence homology to genomic or transcriptomic data, the authors assign the presence of a specific retinoic acid network component to a specific species. The authors then go on to review the functional evidence supporting the role of retinoic acid signaling in the life cycle of echinoderms.

This review is important from an evolutionary stand point to understand the establishment of retinoic acid signaling as a fully functional pathway. I have several general comments on the manuscript that require attention before the manuscript can be accepted for publication.

  1. As researchers working in echinoderms, I can understand the authors want to convince the reader that retinoic acid signaling is functional in these species but the lack of a critical discussion is lacking. An example would be the conservation in echinoderms of an aldh1a gene. First, the gene nomenclature gets extremely confusing. Comparing the aldh1a genes from vertebrates to sequences/genes named raldha-c, or raldh1a1-3, or raldh2, or raldh2a-b, or other combinations make it extremely difficult to reach conclusions.

> We appreciate this author’s comment. Even though the gene names aldh1a1 and aldh2 are correct, we noticed the mistake of using raldh1a1and raldh2 in the diagram of phylogenic tree. We corrected this mistake in the revised manuscript. Furthermore, we modified the gene nomenclature of echinoderm raldhs to clarify the gene orthologues throughout the revised manuscript: raldha, -b and -caldh1a-a, -b and -c.

  1. The issue of incomplete presentation continues from aldh1a to aldh8a1. The identification of these gene seems “solve” the problem. Again, the authors do not cite the work supporting the possible role of this enzyme in retinoic acid biosynthesis (PMID: 11007799) or the more recent paper disputing this contention (PMID: 29703752).

> According to this comment, we modified our manuscript with citing the papers which this reviewer kindly suggested.

Thus, no aldh1a gene has been retrieved from the genomes of these species, although Campo‐Paysaa et al. [7] identified sea urchin aldh8a1 genes from genomic data [7]. We also identified aldh8a1 genes from transcriptome data of other echinoderms, including the sea cucumber (Fig. 3), suggesting that RA can be synthesized without aldh1a genes in echinozoans. Therefore, although echinozoans lost the aldh1a gene secondarily, it is likely that RA signaling still functions in this taxon.

To

Aldh8a1 is known as another raldh gene in vertebrates [24]. Although no aldh1a gene has been retrieved from the genomes of sea urchins and sea cucumbers, Campo‐Paysaa et al. [7] identified sea urchin aldh8a1 genes from genomic data [7]. We also identified aldh8a1 genes from other echinoderms, including the sea cucumber (Fig. 4). Although this finding suggests that RA can be synthesized without aldh1a genes in echinozoans, it must be noted that recent work suggest a reassignment of aldh8a1 function to the kynurenine pathway in tryptophan catabolism [25]. … (L110–118).

  1. The fact that no actual enzymatic activity in echinoderms of the enzyme groups is hardly discussed detracts from their identification. The data is all presented as sequence phylogenetic trees and the reader would greatly benefit from some protein/peptide alignments of functional domains that would define specific enzyme sub-families.

> Among RA signaling genes, the functional domains have been comparatively well investigated in the RA synthesis enzyme (aldh1a), degradation enzyme (cyp16) and receptor (rar). Thus, according to this reviewer, we discussed the enzymatic activity with showing their alignments in the supplementary files of the revised manuscript.

Ex) aldh1a

By contrast, we previously identified three aldh1a genes (aldh1a-a, b, and c) from tran-scriptome data of the starfish Patiria pectinifera and one aldh1a gene from transcriptome data of the sea lily Metacrinus rotundus [9,11]. We also identified one aldh1a gene from a genomic dataset of the brittle star Amphiura filiformis (Fig. 2), indicating that common ancestors of echinoderms had aldh1a gene(s). We furthermore found that functional domain residues like NAD binding sites were well conserved among vertebrate and echinoderm aldh1a genes (Fig. S1). (L102–104).

  1. It is also important to clarify when describing functional data whether experiments are based on gain- or loss-of-function.

> According to this and another reviewer’s comments, we modified to clarify the gain- or loss-of-function in the revised manuscript. The example is the following.

Exogenous RA treatment induced metamorphosis in starfish larvae at the competent stage for metamorphosis [9]. By contrast, inhibitors of RA synthesis and RA receptors suppressed metamorphosis triggered by attachment to a substrate via brachiolar arms.

to

In the gain-of-function experiment, exogenous RA treatment induced metamorphosis in starfish larvae at the competent stage for metamorphosis [9]. By contrast, the loss-of-function by inhibitors of RA synthesis and RA receptors suppressed metamorphosis triggered by attachment to a substrate via brachiolar arms (L250–257).

  1. An even more general comment is the zoological descriptions. This reviewer has some zoological and evolutionary knowledge. When only species names are given but no general names like starfish or sea urchin are mentioned, it required me to search for the species to know what the authors were talking about. It is very important to make the manuscript readable to as general an audience as possible.

> We modified our manuscript to add the general names throughout the revised manuscript.

  1. A review is a great format to summarize data and present hypothesis while refuting others. This review does a good sequence phylogenetic comparisons but lacks in the functional aspects of the genes/transcripts identified. Perhaps a summary table of sea urchins vs starfish vs sea cucumbers, etc. and which genes they have and which don't would give a clearer picture of the conservation of the retinoic acid network in echinoderms

> According to this comment, we added a summary figure of gene identification (Fig. 8) in the revised manuscript.

  1. The manuscript should be proofread to improve the English.

> Modified Accordingly.

Reviewer 3 Report

I found this to be a very interesting and useful summary of RA signaling in echinoderms and other invertebrates and the new molecular phylogenetic analyses are valuable. I have only two major recommendations and a few minor suggestions concerning the writing.

Major revisions:

1)  A table should be included that indicates the specific versions of the genome and transcriptome assemblies used for the molecular phylogenetic analysis, and the quality of these assemblies needs to be evaluated at appropriate places in the text. For example, it was not clear whether the newest and most complete assembly of the S. purpuratus genome (v.5, available both at NCBI and Echinobase) was used in this study. It is important that the authors use this assembly. When the authors state (L 65): “Notably, no aldh1a genes were retrieved from transcriptome or genomic data of the sea cucumber Apostichopus japonicus, suggesting that aldh1a was lost in common ancestors of echinozoans (sea urchins and sea cucumbers),” and (L72) “echinozoans lost the aldh1a gene secondarily…”, this claim is not well supported if high-quality genome assembles were not available (for the sea cucumber) or not used (for the sea urchin). The quality of the various genomic data used in the analysis is crucial and needs to be discussed in the paper and the datasets used should be identified more clearly.

2) The analysis of RAREs should be clarified and expanded (L161-167). It is stated that “The present genomic survey revealed that RAREs position in the promoter regions of RA signaling-related genes such as chordate Hox genes” but data need to included that support this statement. More generally, how prevalent is are RARE sequences in the highest-quality echinoderms genomes? What fraction of genes have a consensus RARE element near the TSS? 

Minor revisions:

1) Statements regarding the role of RA signaling in starfish metamorphosis are contradictory. On L32 the authors state: “Previously, we reported that RA signaling mediates the metamorphosis process in two starfish species”.  Yet on L261 they state: “The role of RA signaling in starfish remains an open question…”. Based on their published work, I don’t think the second sentence is intended as written and should probably be deleted. Here they are not questioning whether RA signaling has any role, but what mechanism(s) might delay the timing of signaling even though the genes are expressed.

2) L24 “a genomic survey” (singular) revealed that non-chordate invertebrates also possess the components, but cite 3 references here [5–7]. Are they referring to one of these 3 studies in particular? If they want to point to all 3 studies, better to say that “several comparative genomic studies have shown that…” or something similar.

3) L30-31- “highly specified penta-radial body plan”- Replace “highly specified” with “distinctive”, “unique”, “characteristic” or something similar.

4) L58- “The aldh1a genes are focused on the RA signaling pathway”- It’s not clear what “focused on” means here. Do you mean “devoted to”, in the sense that they only function in this pathway? 

5) L136 “In vertebrates, RA functions by binding with the RAR–RXR heterodimer, which usually has three rar genes “ Change last part to “the components of which are encoded by three genes in most vertebrates…”

6) L189- Replace “Several reagent treatment experiments” with “Several studies have examined the effects of exogenous RA on echinoderm developmen”. 

Author Response

Reviewer 3

I found this to be a very interesting and useful summary of RA signaling in echinoderms and other invertebrates and the new molecular phylogenetic analyses are valuable. I have only two major recommendations and a few minor suggestions concerning the writing.

Major revisions:

1)  A table should be included that indicates the specific versions of the genome and transcriptome assemblies used for the molecular phylogenetic analysis, and the quality of these assemblies needs to be evaluated at appropriate places in the text. For example, it was not clear whether the newest and most complete assembly of the S. purpuratus genome (v.5, available both at NCBI and Echinobase) was used in this study. It is important that the authors use this assembly. When the authors state (L 65): “Notably, no aldh1a genes were retrieved from transcriptome or genomic data of the sea cucumber Apostichopus japonicus, suggesting that aldh1a was lost in common ancestors of echinozoans (sea urchins and sea cucumbers),” and (L72) “echinozoans lost the aldh1a gene secondarily…”, this claim is not well supported if high-quality genome assembles were not available (for the sea cucumber) or not used (for the sea urchin). The quality of the various genomic data used in the analysis is crucial and needs to be discussed in the paper and the datasets used should be identified more clearly.

> We added the table which clarifies the datasets used in this study (Table 1). In addition to the S. purpuratus genomic data which we previously used (v.3.1), we examined to identify aldh1a genes using the latest version of S. purpuratus data (v.5.0) in the revised manuscript. For the dataset of A. japonicus, we used the latest version of genomic data with high-quality of assembly. We clarified that any aldh1a genes were not retrieved from even these data with high quality in the revised manuscript as following.

Notably, no aldh1a genes were retrieved from transcriptome or genomic data of the sea cucumber Apostichopus japonicus, suggesting that aldh1a was lost in common ancestors of echinozoans (sea urchins and sea cucumbers).

To

Notably, no aldh1a genes were retrieved from transcriptome or genomic data of the sea cucumber Apostichopus japonicus. Regarding that we explored the latest genome assembly of the sea urchin S. purpuratus (Spur_v5.0, 921 Mb, contig N50: 2 Mb, scaffold N50: 37 Mb) and the sea cucumber A. japonicus (805 Mb, contig N50: 190 Kb, scaffold N50: 486 Kb) to identify the aldh1a genes [14], it is likely that aldh1a was lost in common ancestors of echinozoans (sea urchins and sea cucumbers). (L104–109).

We also want to note another modification. The data from the brittle star Amphiura filiformis was taken as genomic data in the previous manuscript, but the correct data type was transcriptome data. This point has been corrected in the revised manuscript.

2) The analysis of RAREs should be clarified and expanded (L161-167). It is stated that “The present genomic survey revealed that RAREs position in the promoter regions of RA signaling-related genes such as chordate Hox genes” but data need to included that support this statement. More generally, how prevalent is are RARE sequences in the highest-quality echinoderms genomes? What fraction of genes have a consensus RARE element near the TSS?

> Although we mentioned that “the present genomic survey revealed that …” in the previous manuscripts, this survey was not conducted by us. In order to clarify that we reviewed the previous reports about RARE of vertebrates, we stated that “the recent genomic survey revealed that …” in the revised manuscript (L211). Additionally, we modified the section about RARE to show the details of RARE and findings about RARE of echinoderms in the revised manuscript as following.

L206–218

2.5. Regulation of downstream genes

The RAR–RXR heterodimer recognizes a specific DNA sequence, the RA response element (RARE), in the promoter or enhancer regions of downstream genes [1,34,35]. RARE is typically composed of two direct repeats (DRs) with a conserved nucleotide sequence (A/G)G(G/T)TCA [36]. This sequence is further separated by a spacer of one, two or five nucleotides (respective elements are termed as DR1, DR2, and DR5 elements) [36]. The recent genomic survey revealed that RAREs position in the promoter regions of the downstream genes of RA signaling like Hox genes in chordates [1].

In echinoderms, Marlétaz et al. [3] found a putative DR5 RARE at the 3,770 bp 5’-upstream of the sea urchin hox1 gene [3]. Carvalho et al. [36] also identified the D2 RARE (AGTTCAATAGTTCA) at the 2,503 bp 5’-upstream of the sea urchin cyp26 gene [36]. Despite these reports, the presence of RAREs have not investigated in other echi-noderm genomic data. Furthermore, it is still unclear if RAR/RXR heterodimer specifi-cally recognizes the RARE in the genome of echinoderm.

Minor revisions:

1) Statements regarding the role of RA signaling in starfish metamorphosis are contradictory. On L32 the authors state: “Previously, we reported that RA signaling mediates the metamorphosis process in two starfish species”. Yet on L261 they state: “The role of RA signaling in starfish remains an open question…”. Based on their published work, I don’t think the second sentence is intended as written and should probably be deleted. Here they are not questioning whether RA signaling has any role, but what mechanism(s) might delay the timing of signaling even though the genes are expressed.

> This is also a mistake which occurred during proofreading process for the English. We deleted this sentence from the revised manuscript.

2) L24 “a genomic survey” (singular) revealed that non-chordate invertebrates also possess the components, but cite 3 references here [5–7]. Are they referring to one of these 3 studies in particular? If they want to point to all 3 studies, better to say that “several comparative genomic studies have shown that…” or something similar.

> According to this comment, we modified our manuscript as following.

“however, a genomic survey revealed that non-chordate invertebrates also possess the components of RA signaling”

to

“however, several comparative genomic studies have shown that non-chordate invertebrates also possess the components of RA signaling” (L24–25)

3) L30-31- “highly specified penta-radial body plan”- Replace “highly specified” with “distinctive”, “unique”, “characteristic” or something similar.

> According to this comment, we modified our manuscript as following.

“… have a highly specified penta-radial body plan.”

To

“... have a unique penta-radial body plan.” (L31)

4) L58- “The aldh1a genes are focused on the RA signaling pathway”- It’s not clear what “focused on” means here. Do you mean “devoted to”, in the sense that they only function in this pathway?

> We modified our manuscript to clarify that raldh1–3(aldh1a1–3) has been focused on the research of RA signaling pathway (L96–97).

5) L136 “In vertebrates, RA functions by binding with the RAR–RXR heterodimer, which usually has three rar genes “ Change last part to “the components of which are encoded by three genes in most vertebrates…”

> We modified this sentence as following.

In vertebrates, RA functions by binding with the RAR–RXR heterodimer, which usually has three rar genes (rarα, rarβ, and rarγ) and three rxr genes (rxrα, rxrβ, and rxrγ)

To

“RA functions by binding with the RAR–RXR heterodimer, the components of which are encoded by three genes in most vertebrates” (L171–172).

6) L189- Replace “Several reagent treatment experiments” with “Several studies have examined the effects of exogenous RA on echinoderm developmen”.

> Modified accordingly.

Reviewer 4 Report

Vitamin A (retinol) and its derivatives play several essential roles during embryogenesis and throughout adult life in Metazoa, acting both at the genetic and epigenetic levels. The earliest requirement of atRA in the development of vertebrate deuterostomes occurs in the posteriorization of the body axis, which affects the patterning of several organs, including the spinal cord, forelimbs, heart, eye, and reproductive tracts. Although the role of atRA has been extensively studied in vertebrates, to date very little is known about its function in invertebrates.

In this context, the review of Yamakava and Wada focusing on the potential role of RA signaling in echinoderms will expand our knowledge on RA signaling in invertebrates.

Interestingly, in their manuscript, the authors reported analyses on the conservation of key genes involved in the RA machinery toolkit: synthesis, degradation, transport, and signaling transduction (receptors) within the echinoderms. Furthermore, they then focused on the potential role of RA signaling during the metamorphosis process in echinoderms and discuss the potential role of this molecule during life cycle regulation in common ancestors of cnidarians and bilaterians.

However, although I support the publication in the cells Journal, I think that the author should better clarify some points and expand their manuscript with the suggestions below:

Figure 1: To better complement the figure I suggest adding in it a little bar representing the regulatory elements recognized from the PPAR-RXR heterodimers (PPRE, Peroxisome proliferator response elements), and one for the regulatory elements recognized from the heterodimers RAR-RXR (RARE, retinoic acid response elements). Furthermore, I think that it will be useful to add this information also in the text at lane 26 for the RARE and at lane 46 for the PPRE.

Paragraph 2.1 Lane 56

RA synthesis: While in figure 1 the authors considered the retinol dehydrogenase as the first gene involved in RA synthesis they do not consider it in paragraph 2.1 and focused their attention only on aldh genes. For example, the authors reported the absence of Aldh1a in Strongylocentrotus purpuratus and Apostichopus japonicas. Therefore, given the importance of the rate-limiting enzymes necessary for the dehydrogenation of retinol in retinal (SDR-RDH and MDR-ADH) it would be interesting to expand their genomic survey and phylogenetic analyses also for these genes involved in this reversible reaction within the Echinodermata phylum.

Lane 99-101

The name of the species is not in italic as they have correctly been reported in the other part of the manuscript. Please check it also through the whole manuscript.

 Lane 145-150

The authors describe the RAR-RXR heterodimers of Platynereis dumerlii and Priapulus caudatus as able to activate the expression of downstream genes. The authors should change this sentence and mention that a chimeric construct with the LBD domain of each species fused to the GAL4 domain is able to activate a reporter gene in vitro in presence of RA rather than to activate a downstream gene.

Furthermore, Handberg –Thorsager et al. 2018 mention that the RAR of sea urchin is characterized by a ligand affinity similar to that of PduRAR. The author should mention this.

Furthermore, in relation to the RAR affinity to RA, the authors should mention that there are differences between the data of Handberg –Thorsager et al. 2018 in Platynereis dumerlii where RAR is considered to be a low RA sensor with respect to the data coming from Fonseca et al. 2019 in Priapulus caudatus. Indeed, these two receptors do not function in a similar way as 10 µM of atRA activates Priapulus RAR with 1.7 fold increase compared to the vehicle, whilst 10 µM of atRA is able to activate Platynereis RAR over 140 fold increase. Therefore, I think that it will be useful to discuss that these two receptors could be considered respectively as low sensor and very low sensor.

Figure 6:

The authors should provide a picture with a better resolution.

Paragraph 2.4

The authors performed a very detailed phylogenetic analysis on the key genes of RA machinery. However, it would be interesting to show an alignment of the LBD domain of the different echinoderms RARs object of this review in comparison to one of the fully active vertebrate receptors. These analyses will help to show the levels of identity but also understand how many of the main 25 amino acids known to interact with all-trans RA in their ligand-binding pocket are conserved.  These alignments will definitively contribute to predicting the potential ability of Echinodermata RARs to bind RA and in which way.

Paragraph 2.5

The authors mention that retinoic acid response elements (RARE) have been previously identified in the enhancer region of hox and cyp26 genes in sea urchin and that the same has never been investigated in other echinoderms species. This paragraph is written in a confusing way. Please rewrite it in a more linear way.

Line 195

As cyp26 genes play a key role in RA feedback mechanisms it should be added among the others for the re-examination of the expression analyses.

Line 212 and 235

The authors should specify that is the binding of a RAR antagonist (RO) induces suppression of the metamorphosis in Atropecten latespinosus and Antedon serrata.

Line 224

The authors should mention that in ascidians RA has been shown to be directly involved in the formation of the otic/atrial placode during the metamorphosis as reported in the Development paper of Sasakura Y.  et al 2012.

Author Response

Reviewer 4

Vitamin A (retinol) and its derivatives play several essential roles during embryogenesis and throughout adult life in Metazoa, acting both at the genetic and epigenetic levels. The earliest requirement of atRA in the development of vertebrate deuterostomes occurs in the posteriorization of the body axis, which affects the patterning of several organs, including the spinal cord, forelimbs, heart, eye, and reproductive tracts. Although the role of atRA has been extensively studied in vertebrates, to date very little is known about its function in invertebrates.

In this context, the review of Yamakava and Wada focusing on the potential role of RA signaling in echinoderms will expand our knowledge on RA signaling in invertebrates.

Interestingly, in their manuscript, the authors reported analyses on the conservation of key genes involved in the RA machinery toolkit: synthesis, degradation, transport, and signaling transduction (receptors) within the echinoderms. Furthermore, they then focused on the potential role of RA signaling during the metamorphosis process in echinoderms and discuss the potential role of this molecule during life cycle regulation in common ancestors of cnidarians and bilaterians.

However, although I support the publication in the cells Journal, I think that the author should better clarify some points and expand their manuscript with the suggestions below:

Figure 1: To better complement the figure I suggest adding in it a little bar representing the regulatory elements recognized from the PPAR-RXR heterodimers (PPRE, Peroxisome proliferator response elements), and one for the regulatory elements recognized from the heterodimers RAR-RXR (RARE, retinoic acid response elements). Furthermore, I think that it will be useful to add this information also in the text at lane 26 for the RARE and at lane 46 for the PPRE.

> According to this comment, we modified the figure 1 and the text in our manuscript. Especially, we added the following sentences.

RAR and RXR form a heterodimer that regulates downstream genes by binding to particular DNA sequences (RARE; retinoic acid response elements) (L19–20).

PPAR makes heterodimer with RXR and binds to the specific genomic elements (Fig. 1; PPRE, peroxisome proliferator response elements). (L49–50).

Paragraph 2.1 Lane 56

RA synthesis: While in figure 1 the authors considered the retinol dehydrogenase as the first gene involved in RA synthesis they do not consider it in paragraph 2.1 and focused their attention only on aldh genes. For example, the authors reported the absence of Aldh1a in Strongylocentrotus purpuratus and Apostichopus japonicas. Therefore, given the importance of the rate-limiting enzymes necessary for the dehydrogenation of retinol in retinal (SDR-RDH and MDR-ADH) it would be interesting to expand their genomic survey and phylogenetic analyses also for these genes involved in this reversible reaction within the Echinodermata phylum.

> According to this comment, we added the review about the retinol dehydrogenase with phylogenic analysis in the revised manuscript. Especially, we identified the representative gene dhrs7 and adh3 for SDR-RDH and MDR-ADH in the genomic data of echinoderms, respectively. The followings are modified parts in the revised manuscript.

In vertebrates, retinal is catalyzed to retinal with the usage of retinal dehydrogenase (Fig. 1; RDH) [1,2]. (L42–43)

2.1.1 Retinol to retinal

In vertebrates, RDH genes are grouped into two gene families: SDR-RDH (short-chain dehydrogenase/reductase-RDH, e.g. rdh10/sdr16c4, rdh11/sdr7c1 and dhrs7/retsdr4) and MDR-ADH (medium-chain alcohol dehydrogenase, e.g. adh3, adh4) [6]. Both SDR-RDH and MDR-ADH genes have been identified in non-chordate deu-terostomes, protostomes and even cnidarians [6,18,19], suggesting that the origin of RDH genes predates before the divergence of cnidarians and bilaterians. For example, a recent study identified the dhrs7 genes in sea cucumbers and sea urchins [18], and adh3 genes in protostomes such as annelids [19]. In addition to these identifications, we were able to extract the dhrs7 and adh3 genes from all classes of echinoderms (Fig. 2). This result in-dicates the presence of RDH genes in echinoderms, suggesting that the pathway to cat-alyze retinol to retinal is conserved. Future studies should focus on the comprehensive identification of SDR-RDH and MDR-ADH genes in echinoderms and their ability of retinol oxidation. (L67–79)

Lane 99-101

The name of the species is not in italic as they have correctly been reported in the other part of the manuscript. Please check it also through the whole manuscript.

> Modified accordingly.

Lane 145-150

The authors describe the RAR-RXR heterodimers of Platynereis dumerlii and Priapulus caudatus as able to activate the expression of downstream genes. The authors should change this sentence and mention that a chimeric construct with the LBD domain of each species fused to the GAL4 domain is able to activate a reporter gene in vitro in presence of RA rather than to activate a downstream gene.

> According to this comment, we modified the review about the RA receptors about Platynereis dumerlii and Priapulus caudatus as following.

Insight into the conservation of the RAR–RXR heterodimer can be obtained from protostomes. Handberg-Thorsager et al. [27] reported that RAR forms a heterodimer with RXR to bind all-trans RA in the annelid Platynereis dumerilii [27], and Fonseca et al. [28] demonstrated that the RAR–RXR heterodimer activates the expression of downstream genes in the priapulid Priapulus caudatus [28].

To

Insight into the conservation of the RA receptor function can be obtained from protostomes. Handberg-Thorsager et al. [19] reported that RAR forms a heterodimer with RXR to bind all-trans RA in the annelid Platynereis dumerilii [19]. They also revealed that a chimeric construct with the ligand binding domain (LBD) of RAR fused to the GAL4 domain is able to activate a reporter gene in vitro in presence of RA [19]. Moreover, Fonseca et al. [33] demonstrated that the LBD of RAR can induce a reporter gene in the priapulid Priapulus caudatus [33].

Furthermore, Handberg –Thorsager et al. 2018 mention that the RAR of sea urchin is characterized by a ligand affinity similar to that of PduRAR. The author should mention this.

> According to this comment, we mentioned about the previous discussion about a ligand affinity of sea urchin RAR function as following.

Handberg-Thorsager et al. [27] discussed that the ligand binding affinity of sea urchin RAR is comparable to the annelid P. dumerilii RAR. (L193–194)

Furthermore, in relation to the RAR affinity to RA, the authors should mention that there are differences between the data of Handberg–Thorsager et al. 2018 in Platynereis dumerlii where RAR is considered to be a low RA sensor with respect to the data coming from Fonseca et al. 2019 in Priapulus caudatus. Indeed, these two receptors do not function in a similar way as 10 µM of atRA activates Priapulus RAR with 1.7 fold increase compared to the vehicle, whilst 10 µM of atRA is able to activate Platynereis RAR over 140 fold increase. Therefore, I think that it will be useful to discuss that these two receptors could be considered respectively as low sensor and very low sensor.

> According to this comment, we added the following discussion about the low-affinity of ligand binding in the revised manuscript.

The two studies mentioned above also shed light on the affinity of ligand binding of RAR. Handberg-Thorsager et al. [19] revealed that the RAR of the annelid P. dumerilii shows a lower ligand binding affinity than the RAR of chordates [19]. Fonseca et al. [33] found that the ligand binding affinity of RAR is much lower in the priapulid P. caudatus than in the annelid P. dumerilii [33]. These reports suggested that the ancestral RAR is low-affinity sensor for RA.

Figure 6:

The authors should provide a picture with a better resolution.

> Modified accordingly.

Paragraph 2.4

The authors performed a very detailed phylogenetic analysis on the key genes of RA machinery. However, it would be interesting to show an alignment of the LBD domain of the different echinoderms RARs object of this review in comparison to one of the fully active vertebrate receptors. These analyses will help to show the levels of identity but also understand how many of the main 25 amino acids known to interact with all-trans RA in their ligand-binding pocket are conserved. These alignments will definitively contribute to predicting the potential ability of Echinodermata RARs to bind RA and in which way.

> According to this comment, we investigated the alignment of echinoderm RARs and human RARs to reveal how many of the main 25 amino acids known to interact with all-trans RA in their ligand-binding pocket are conserved. As results, we revealed several mismatch-residue of ligand binding pocket. We discussed this in relation with the ligand binding affinity in the revised manuscript as following.

Handberg-Thorsager et al. [19] discussed that the ligand binding affinity of sea urchin RAR is comparable to the annelid P. dumerilii RAR [19]. In addition to these reports, this study also found that the echinoderm RAR showed several mismatch-residue in their ligand binding pockets with the human RAR (Fig. S3). It is interesting to determine the ligand binding affinity of echinoderm RARs. (L193–197).

Paragraph 2.5

The authors mention that retinoic acid response elements (RARE) have been previously identified in the enhancer region of hox and cyp26 genes in sea urchin and that the same has never been investigated in other echinoderms species. This paragraph is written in a confusing way. Please rewrite it in a more linear way.

> According to this and another comments, we modified as following.

…Marlétaz et al. [3] and Carvalho et al. [31] each found a putative RARE in the promoter region of sea urchin Hox genes and the cyp26 gene [3,31], although no further functional studies have been conducted. To date, no study has comprehensively investigated RAREs using echinoderm genomic data.

to

In echinoderms, Marlétaz et al. [3] found a putative DR5 RARE at the 3,770 bp 5’-upstream of the sea urchin hox1 gene [3]. Carvalho et al. [36] also identified the D2 RARE (AGTTCAATAGTTCA) at the 2,503 bp 5’-upstream of the sea urchin cyp26 gene [36]. Despite these reports, the presence of RAREs have not investigated in other echi-noderm genomic data. Furthermore, it is still unclear if RAR/RXR heterodimer specifi-cally recognizes the RARE in the genome of echinoderm. (L213–218).

Line 195

As cyp26 genes play a key role in RA feedback mechanisms it should be added among the others for the re-examination of the expression analyses.

> According to this comment, we modified as following.

“However, these results should be re-examined using expression analyses of signaling component genes such aldh1a and receptors rar and rxr.”

To

“However, these results should be re-examined using expression analyses of signaling component genes such aldh1a, rar, rxr and cyp26.” (L251–252).

Line 212 and 235

The authors should specify that is the binding of a RAR antagonist (RO) induces suppression of the metamorphosis in Atropecten latespinosus and Antedon serrata.

> According to this reviewer, we modified the manuscript as following.

…, the inhibition of RA synthesis or binding to RAR suppressed metamorphosis,…

To

…, the inhibition of RA synthesis or the binding of a RAR antagonist (RO41-5253) suppressed metamorphosis…” (L269–270).

Line 224

The authors should mention that in ascidians RA has been shown to be directly involved in the formation of the otic/atrial placode during the metamorphosis as reported in the Development paper of Sasakura Y. et al 2012.

> According to this comment, we modified the manuscript to mention about the work by Sasakura et al., 2012 as following.

RA signaling has not been reported to be involved in the metamorphosis of sea urchins or ascidians [36,37], deuterostomes among which molecular metamorphosis regulation has been investigated in detail.

To

In the deuterostomes, Sasakura et al. 2012 reported that RA signaling mediates the ex-pression of hox1 which is required for the epidermal otic/atrial placodes formation during metamorphosis of the ascidian Ciona intestinalis [41]. However, RA signaling has not been reported to be involved in the metamorphosis regulation of sea urchins or ascidians [42,43], deuterostomes among which molecular metamorphosis regulation has been investigated in detail. (L301–305).
